# Impact of impurities in bromocresol green indicator dye on spectrophotometric total alkalinity measurements

Katharina Seelmann[1], Martha Gledhill[1], Steffen Aßmann[2], and Arne Körtzinger[1,3]

[1]GEOMAR Helmholtz Centre for Ocean Research Kiel, Kiel, Germany
[2]Kongsberg Maritime Contros GmbH, Kiel, Germany (at the time of this study)
[3]Christian-Albrechts-Universität zu Kiel, Kiel, Germany

**Correspondence:** Katharina Seelmann (kseelmann@geomar.de)

**Abstract.** Due to its accurate and precise character, the spectrophotometric pH detection is a common technique applied in measurement methods for carbonate system parameters. However, impurities in the used pH indicator dyes can influence the measurements quality. During our work described here, we focused on impacts of impurities in the pH indicator dye bromocresol green (BCG) on spectrophotometric seawater total alkalinity ($A_\mathrm{T}$) measurements. In order to evaluate the extent of such influences, purified BCG served as a reference. First, a high-performance liquid chromatography (HPLC) purification method for BCG was developed as such a method did not exist at the time of this study. An analysis of BCG dye from four different vendors with this method revealed different types and quantities of impurities. After successful purification, $A_\mathrm{T}$ measurements with purified and unpurified BCG were carried out using the novel autonomous analyzer CONTROS HydroFIA® TA. Long-term measurements in the laboratory revealed a direct influence of impurity types and quantities on the drift behavior of the analyzer. The purer the BCG, the smaller was the $A_\mathrm{T}$ increase per measurement. The observed drift is generally caused by deposits in the optical pathway mainly generated by the impurities. However, the analyzers drift behavior could not be fully overcome. Furthermore, we could show that a certain impurity type in some indicator dyes changed the drift pattern from linear to non-linear, which can impair long-term deployments of the system. Consequently, such indicators are impractical for these applications. Laboratory performance characterization experiments revealed no improvement of the measurement quality (precision and bias) by using purified BCG as long as the impurities of the unpurified dye do not exceed a quantity of 2 % (relationship of peak areas in the chromatogram). However, BCG with impurity quantities higher than 6 % provided $A_\mathrm{T}$ values, which failed fundamental quality requirements. Concluding, to gain optimal $A_\mathrm{T}$ measurements especially during long-term deployments, an indicator purification is not necessarily required as long as the purchased dye has a purity level of at least 98 % and is free of the previously named impurity type. Consequently, high-quality $A_\mathrm{T}$ measurements do not require pure but the purest BCG that is purchasable.

## 1 Introduction

Global observations of the marine carbonate system are of high importance to understand biogeochemical processes in the ocean effected by anthropogenic $CO_2$. The measurable key variables characterizing the ocean carbon cycle are pH, total alkalinity ($A_\mathrm{T}$), $pCO_2$, and total dissolved inorganic carbon ($C_\mathrm{T}$). Due to their thermodynamic relationships, it is only necessary to

measure two of these four parameters for a full characterization of the marine carbonate system (Millero, 2007). Traditionally, $A_T$ and $C_T$ were the preferred parameters for this purpose when measuring discrete samples. However, more recently, pH measurements have become more prominent within the oceanographic communities. During decades of ocean carbon observations, several analytical methods have been established, ranging from manual bench top systems for laboratory work via at-sea flow-through analyzers to in situ sensors. Among all these available methods, spectrophotometric pH determination techniques using sulphonephthalein indicator dyes are described as simple, fast, and precise (e.g. Clayton and Byrne, 1993; Tapp et al., 2000; Bellerby et al., 2002; Aßmann et al., 2011). They have been utilized in marine research especially for ocean carbon observations since the late 1980's (Robert-Baldo et al., 1985; Byrne, 1987; Byrne and Breland, 1989; King and Kester, 1989). Since Breland and Byrne (1993) showed that the sulphonephthalein indicator dye bromocresol green (BCG) is suitable for seawater pH determination in the pH range 3.4 to 4.6, it has been used in several spectrophotometric $A_T$ measurement systems with comparable precision and accuracy as traditional methods (Yao and Byrne, 1998; Li et al., 2013; Seelmann et al., 2019).

Investigations of Yao et al. (2007) on seawater pH measurements with the most common indicator dye, meta-cresol purple (mCP) from different vendors have revealed different types and quantities of light-absorbing impurities. These impurities can contribute to pH offsets of up to 0.01 pH units. To overcome the uncertainties caused by indicator impurities, Liu et al. (2011) developed a preparative high-performance liquid chromatography (HPLC) method to purify mCP and characterized this purified dye. Furthermore, to produce large batches of purified mCP, Patsavas et al. (2013a) developed a flash chromatography (FC) method resulting in a 3.5 times increased yield per run. However, not all users of spectrophotometric seawater pH measurement systems are able to purify or to purchase purified mCP. Therefore, Douglas and Byrne (2017) published a mathematical correction for accurate pH measurements using unpurified mCP.

In order to apply these findings to spectrophotometric $A_T$ measurements, Nand and Ellwood (2018) described a simple colorimetric method for determining seawater $A_T$ using purified bromophenol blue (BPB) as pH indicator dye. However, at the time of this study, there are no comparable detailed investigations on how indicator impurities in BCG may influence spectrophotometric seawater $A_T$ measurements.

Since our previous work dealt with an open-cell single-point titration analyzer with spectrophotometric pH determination using BCG as indicator dye (Seelmann et al., 2019), we investigated the influences of any impurities in BCG from different vendors in comparison to purified BCG as reference. Hence, the first step of this study was to develop a purification method for BCG. Due to similarity in the chemical structure of BCG and mCP (see Fig. 1) and the available facilities in our laboratory, we decided to develop an HPLC analysis and purification method for BCG based on the mCP purification method published by Liu et al. (2011). Once the developed method was sufficient for BCG purification, a small batch of purified BCG was produced. Following this, comparative experiments were carried out with a novel autonomous analyzer for seawater $A_T$ using purified and unpurified BCG in order to evaluate the influence of impurities in the indicator dye.

**Figure 1.** Chemical structure of bromocresol green and meta-cresol purple

## 2  Materials and methods

### 2.1  HPLC method

#### 2.1.1  Reagents and instrumentation

The BCG indicator (as sodium salt) was obtained from the following vendors: Acros Organics, Alfa Aesar, Carl Roth, and TCI.
The solvents used in the HPLC purification were water ($H_2O$), acetonitrile (ACN), and trifluoroacetic acid (TFA). The ACN (HPLC grade) and the TFA (Purity: $\geq 99.9$ %) were obtained from Fisher Scientific and Carl Roth, respectively.

A Shimadzu liquid chromatography (LC) system performed both the analytical and preparative chromatography. This system included an auto-sampler (SIL-10ADvp) (only for analytical mode), a isocratic preparative LC pump (LC-8A), an isocratic analytical HPLC pump (LC-10ADvp), a column oven (CTO-10ASvp), a single channel UV-VIS detector (SPD-10Avp), and
an LC controller (SCL-10Avp).

The Primesep B2 HPLC columns were obtained from SIELC Technologies. This is a reverse-phase column with embedded basic ion-pairing groups that retains analytes by reverse-phase and ion-exchange mechanisms. For developing the purification method, an analytical Primesep B2 column (4.6 x 250 mm, particle size: 5 µm) was chosen. The purification was performed by a preparative Primesep B2 column (21.2 x 250 mm, particle size: 5 µm). Analytical separations were performed at 25 °C,
but preparative chromatography was undertaken at room temperature.

#### 2.1.2  Method development

The method development included the optimization of the mobile phase composition for BCG separation on the chosen column and was performed in analytical mode. For this purpose, a 10 mg mL$^{-1}$ BCG solution of each vendor was prepared in the mobile phase and 20 µL were injected. One HPLC run with a flow-rate of 1.5 mL min$^{-1}$ took 60 min and was monitored
using the UV-VIS detector at 280 nm. The optimal mobile phase composition was determined by systematically changing the concentrations of the solvents starting from the conditions described by Liu et al. (2011). There, the mobile phase composition

was 70:30 ACN:H$_2$O (volume:volume) with 0.05 % of TFA. Afterwards, the ACN and TFA concentration were increased by 5 % and 0.05 % increments, respectively, until the mobile phase consisted of 85 % ACN and 0.2 % TFA. One BCG injection was done per mobile phase composition each followed by a blank run. Blank runs were carried out by injecting the mobile

phase as sample.

### 2.1.3    Comparison of BCG from different vendors

Once the optimal mobile phase composition was found, we tested BCG from different vendors for impurity types and quantities. For that, a BCG solution of each vendor was prepared and analyzed as described in Sect. 2.1.2 with the optimal mobile phase composition. To quantitatively compare the purity of each dye, we defined the relative purity of BCG at 280 nm wavelength

($P_{\mathrm{BCG}}$), which was calculated as follows:

$$P_{\mathrm{BCG}} = \frac{A_{\mathrm{BCG}}}{\sum_{\mathrm{i}=1}^{n} A_{\mathrm{i}}} \times 100\% \tag{1}$$

where $A_{\mathrm{BCG}}$ is the area of the BCG peak, $n$ is the number of peaks, and $A_{\mathrm{i}}$ is the area of the i$^{\mathrm{th}}$ peak.

### 2.1.4    Purification of BCG

The purification was performed by the LC system in preparative mode. A 7.5 mg mL$^{-1}$ BCG solution was prepared in the

mobile phase and 10 mL were injected onto the preparative column. Impurities were separated by isocratic flow (flow rate 31.2 mL min$^{-1}$) with 75:25:0.1 ACN:H$_2$O:TFA as mobile phase. The pure BCG was collected manually in a round bottom flask at its retention time of about 52 min. Approximately 90 % of the mobile phase was removed from the BCG eluate using a rotary evaporator, with the final 10 % left to evaporate in a dark open box at room temperature. The pure crystalline dye was transferred to a brown flask for further experiments.

In order to verify the success of the purification, the purified BCG was analyzed by the analytical HPLC procedure as described in Sect. 2.1.2.

## 2.2    Total alkalinity measurements

### 2.2.1    Reagents and instrumentation

Total alkalinity measurements were performed using the novel autonomous analyzer CONTROS HydroFIA$^{\circledR}$ TA (Kongsberg

Maritime Contros GmbH, Kiel, Germany). Its measurement principle is based on a single-point open-cell titration of the seawater sample with subsequent spectrophotometric pH detection using BCG as indicator (Breland and Byrne, 1993; Yao and Byrne, 1998; Li et al., 2013; Seelmann et al., 2019). The seawater sample was titrated with 0.1 mol kg$^{-1}$ hydrochloric acid (HCl) obtained from Carl Roth and constantly temperature controlled to $25.0 \pm 0.1$ °C by the systems internal heat exchanger.

The $A_{\mathrm{T}}$ value of the sample was calculated by the following general equation:

$$\frac{-V_{\mathrm{sw}} \times \rho_{\mathrm{sw}} \times A_{\mathrm{T}} + V_{\mathrm{t}} \times \rho_{\mathrm{t}} \times C_{\mathrm{t}}}{V_{\mathrm{sw}} \times \rho_{\mathrm{sw}} + V_{\mathrm{t}} \times \rho_{\mathrm{t}}} = [\mathrm{H}^+]_{\mathrm{F}} + [\mathrm{HF}] + [\mathrm{HSO}_4^-] + [\mathrm{HI}^-] \tag{2}$$

where $V_{\text{sw}}$ and $V_{\text{t}}$ are the volumes of the seawater sample and the added titrant (HCl and BCG solutions), respectively, and $\rho_{\text{sw}}$ and $\rho_{\text{t}}$ are the densities of the seawater sample and the added titrant, respectively. $C_{\text{t}}$ is the acid concentration in the combined titrant solution. $[\text{H}^+]_{\text{F}}$ is the free concentration of hydrogen ions, and $[\text{HI}^-]$ is the concentration of the protonated (i. e. acidic) form of BCG. $[\text{HF}]$ and $[\text{HSO}_4^-]$ are the concentrations of hydrogen fluoride and the bisulfate ion in the seawater sample. $[\text{H}^+]_{\text{F}}$, or $\text{pH}_{\text{F}}$, in the sample-titrant mixture is measured spectrophotometrically. Following Breland and Byrne (1993) and Yao and Byrne (1998), $\text{pH}_{\text{F}}$ is described by

$$\text{pH}_{\text{F}} = 4.4166 + 0.0005946 \times (35 - S_{\text{mix}}) + \log\left(\frac{R - 0.0013}{2.3148 - R \times 0.1299}\right) \tag{3}$$

where $S_{\text{mix}}$ is the salinity of the sample-titrant mixture, and $R$ is the ratio between the absorbances at 444 and 616 nm.

Certified reference material, CRM, (batch 160, $A_{\text{T,reference}} = 2212.44 \pm 0.67\ \mu\text{mol kg}^{-1}$) was obtained from A. G. Dickson at the Scripps Institution of Oceanography of the University of California, San Diego. The seawater for our experiments was prepared by diluting a commercially available 8.33-fold concentrate of seawater ("Absolute Ocean", ATI Aquaristik) with deionized water. Its absolute $A_{\text{T}}$ value was not important as it was only used for mimicking semi-continuous measurement conditions between the references. All total alkalinity measurements were carried out in an air-conditioned laboratory and after the system was "calibrated" with a freshly opened CRM. However, the "calibration" routine conducted by the CONTROS HydroFIA® TA is not a calibration in a true sense. It rather serves the determination of the exact sample volume by utilizing a one point CRM measurement. The seawater sample volume is the only unknown variable of the absolute $A_{\text{T}}$ determination method (Seelmann et al., 2019).

### 2.2.2 Long-term measurements

For the long-term measurements, $0.002\ \text{mol kg}^{-1}$ BCG solutions were prepared from unpurified BCG (from different vendors) and purified BCG and used as indicator dye in the analyzer. The unpurified dyes (sodium salts) were dissolved in deionized water (DI-water). The purified dye was dissolved in DI-water with sodium hydroxide (NaOH) as additive. The exact amount of NaOH was calculated by the molar ratios and molar masses of BCG and NaOH. This transferred the pure BCG to its sodium salt and improved its solubility. For both unpurified and purified BCG solutions the ionic strength was kept very low (only created by the dissolved BCG sodium salt itself) in order to realize high concentrations of BCG stock solution. However, the dilution of the sample seawater by the added BCG and HCl solution was accounted for in the $A_{\text{T}}$ calculation procedure.

The prepared seawater sample ($\approx 25$ L) was measured more than 300 times with a measuring interval of 15 min, which took about four days. For monitoring the drift, a freshly opened CRM was measured at the beginning and at the end of this experiment, as well as daily in between. Each of these CRM measurements consisted of five consecutive single measurements.

### 2.2.3 Standard addition experiment

In order to evaluate the impact of impurities on the measuring performance of the system, we carried out a standard addition experiment with each unpurified and the purified BCG. This experiment is the standard validation procedure for evaluating the performance of the analyzer under laboratory conditions. Therefore, a seawater sample (with relatively high $A_{\text{T}}$) was titrated

with an HCl solution ($0.1 \, \mathrm{mol \, kg^{-1}}$) to lower its $A_T$ in five steps. The titration was carried out by adding different precisely known volumes of HCl to a known volume of seawater resulting in five seawater samples with stepwise decreased $A_T$. The theoretical $A_T$ ($A_{T,\mathrm{titrated}}$) was calculated from the volumes of added acid and seawater, the concentration of the acid, and the original $A_T$ of the seawater. To determine the practical $A_T$ ($A_{T,\mathrm{measured}}$), each of these samples was repeatedly measured with the analyzer ($n = 5$).

The precision was determined by averaging the standard deviation ($\sigma$) of each sample measurement. The root mean square error (RMSE) of the linear regression after plotting $A_{T,\mathrm{measured}}$ vs. $A_{T,\mathrm{titrated}}$ gave us information about the bias of the method. It was calculated by

$$\mathrm{RMSE} = \pm\sqrt{\frac{1}{n} \times \sum_{i=1}^{n}(A_{T,\mathrm{fitted,i}} - A_{T,\mathrm{measured,i}})^2} \tag{4}$$

where $n$ is the number of samples, $A_{T,\mathrm{fitted,i}}$ is the i$^{\mathrm{th}}$ $A_T$ value calculated with the linear regression equation with $A_{T,\mathrm{titrated,i}}$ as $x$ variable, where $A_{T,\mathrm{measured}}$ is the average of the five repeatedly measured $A_T$ values of each titrated seawater sample. Slope and intercept of this regression were important for the evaluation of linearity and sensitivity. Within the standard validation procedure of the analyzer, these terms must fulfill within their uncertainties the following requirements: Slope = 1; intercept = 0.

## 3 Results and discussion

### 3.1 HPLC separation and purification of BCG

#### 3.1.1 Method development

Table 1 summarizes the influence of the different mobile phase compositions on the BCG separation. For saving solvents and time, it is important to keep the time of each HPLC run under 60 min, but, at the same time, with an optimal separation of BCG from its impurities. Hence, the optimal separation of BCG was achieved with 75:25:0.1 ACN:$H_2O$:TFA as mobile phase. The pure BCG was eluted from the column as fast as possible (retention time: 52 min) with the best dye-impurity separation.

#### 3.1.2 Comparison of BCG from different vendors

Figure 2 shows the resulting analytical HPLC chromatograms. There, BCG from different vendors shows different types and quantities of impurities. The retention time of the pure BCG was 52 min in all chromatograms. Another similarity between all chromatograms was the cluster of several peaks around 3-5 min. Only the peak areas of these peaks strongly differed. As there are no peaks at these retention times in the blank chromatogram, this peak cluster had to be caused by the indicator and not by the solvent. BCG from Acros Organics, Alfa Aesar and Carl Roth showed an intensive peak around 58 min, which is not present in the BCG from TCI. However, BCG from TCI showed three other small peaks around 26 min, 29 min, and 42 min. Alfa Aesar BCG also showed the 42 min peak in addition to small peaks around 7 min, 10 min, and 35 min. These various

**Table 1.** Mobile phase compositions and their impact on the BCG separation

| Mobile phase composition: | | | BCG separation: | |
|---|---|---|---|---|
| ACN (%) | H$_2$O (%) | TFA (%) | BCG peak (min) | Sufficient separation of impurities |
| 70 | 30 | 0.05 | no elution[a] | - |
| 70 | 30 | 0.10 | 60 | no[b] |
| 70 | 30 | 0.15 | no elution[a] | - |
| 70 | 30 | 0.20 | no elution[a] | - |
| 75 | 25 | 0.10 | 52 | yes |
| 80 | 20 | 0.10 | 56 | yes |
| 85 | 15 | 0.10 | 60 | no[b] |

[a]Within 60 min run time

[b]Impurities found in subsequent blank run

**Table 2.** Summary of analytical HPLC of unpurified BCG from different vendors

| | Acros Organics | Alfa Aesar | Carl Roth | TCI |
|---|---|---|---|---|
| Number of peaks | 3 | 7 | 3 | 5 |
| $P_{\mathrm{BCG}}$ (%) | 93.2 | 85.4 | 92.5 | 98.1 |

quantities of total absorbance at 280 nm resulted in different $P_{\mathrm{BCG}}$. The calculated $P_{\mathrm{BCG}}$ for each vendor (following Equ. 1) are summarized in Table 2. It has to be taken into account that these quantities are only valid when using an UV detector. Other detectors may result in different purity levels.

### 3.1.3 Purification of BCG

In order to test the effectiveness of the purification method, we decided to purify the least pure BCG from Alfa Aesar. Furthermore, to produce the most pure dye, also BCG from TCI was chosen for purification. The obtained yields were between 60 % and 70 % for both BCGs with around 50 mg purified BCG recovered per injection.

Figure 3 shows the analytical HPLC chromatograms of purified TCI BCG, and Alfa Aesar BCG. Both chromatograms still show the peak cluster around 3-5 min, but with much smaller areas, especially with purified Alfa Aesar BCG. Furthermore, the 42 min and 58 min peaks of Alfa Aesar BCG could not be totally removed. However, the purity of TCI BCG, and Alfa Aesar BCG improved to 99.6 %, and 99.3 %, respectively. The results are summarized in Table 3.

Since the relative purity of Alfa Aesar BCG was improved from 85.4 % to 99.3 %, the success of the purification was proven. Hence, the HPLC purification method developed here is considered sufficient for the nearly full removal of impurities from BCG.

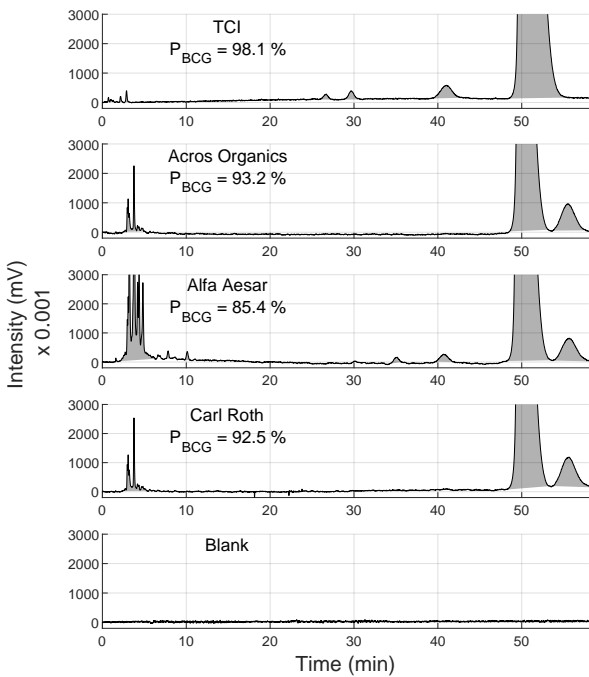

**Figure 2.** Analytical HPLC chromatograms of unpurified BCG from different vendors with their $P_{\mathrm{BCG}}$ and a chromatogram from a solvent injection without BCG (Blank). All peaks are highlighted with gray background color.

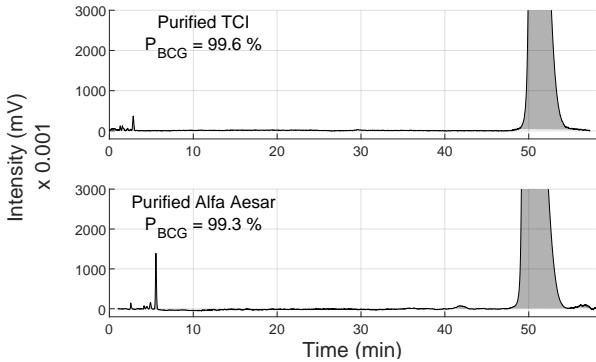

**Figure 3.** Analytical HPLC chromatograms of purified BCG from TCI, and Alfa Aesar with their $P_{\mathrm{BCG}}$. All peaks are highlighted with gray background color.

**Table 3.** Summary of analytical HPLC of purified BCG

|  | Purified from | |
|---|---|---|
|  | TCI | Alfa Aesar |
| Number of peaks | 2 | 4 |
| $P_{\mathrm{BCG}}$ (%) | 99.6 | 99.3 |

## 3.2 Total alkalinity measurements

### 3.2.1 Long-term measurements

During previous studies with the CONTROS HydroFIA® TA analyzer, we found that a linear drift to higher $A_{\mathrm{T}}$ values appears to be the typical behavior of the system (Seelmann et al., 2019). We also found out that the drift is caused by deposits in the optical pathway. As a result, the light intensity decreases and therefore the absorbances at 444 and 616 nm (wavelengths where the acid and base form of BCG have their absorbance maxima) changes in a certain ratio so that the $A_{\mathrm{T}}$ values increases per measurement. In the present study we wanted to examine the impact of BCG impurities and the usage of purified BCG, respectively, on the drift behavior of the system.

In order to evaluate the drift of the system supposedly caused by impurities of the BCG indicator dye, the bias between the measured $A_{\mathrm{T}}$ value and the reference $A_{\mathrm{T}}$ value of the CRM ($\Delta A_{\mathrm{T}} = A_{\mathrm{T,measured}} - A_{\mathrm{T,reference}}$) was plotted vs. the measurement counter. Figure 4 shows the results for $A_{\mathrm{T}}$ measurements with purified and unpurified TCI BCG, as well as unpurified BCG from Alfa Aesar and Acros Organics. Measurements with purified and unpurified TCI BCG resulted in a linear drift to higher values with the regression equation $y = (0.0193 \pm 0.0009) \times x + (-0.18 \pm 0.16)$, and $y = (0.0317 \pm 0.0004) \times x + (-0.16 \pm 0.10)$, respectively. However, unpurified Acros Organics and Alfa Aesar BCG showed a non-linear drift to higher values. All $A_{\mathrm{T}}$ measurements took into account the relative uncertainty of the analyzer, determined as 0.08 % (Seelmann et al., 2019). Figure 4 does not show these uncertainties as they are to small for the scaling of the y-axis.

One important outcome of this experiment is, that the magnitude and shape of the drift directly depends on the purity of the used BCG. The drift caused by purified TCI BCG is reduced by 0.0124 µmol kg$^{-1}$ per measurement with respect to unpurified TCI BCG. This indicates that the drift of the system must be primarily caused by impurities of the BCG indicator and not by the indicator itself as hypothesized in our previous study (Seelmann et al., 2019). However, there is still a remaining small drift component even with the most pure TCI BCG. Hence, BCG purification appears to significantly reduce but not completely eliminate the observed system drift. For resetting the drift, a flush with ethanol or isopropyl alcohol removes any impurity deposits in the optical pathway caused by the indicator dye. The frequency of these cleanings during long-term deployments can be reduced by using purer dye. But finally, the user of the CONTROS HydroFIA® TA analyzer decides the cleaning interval as its frequency depends on the certain application of the system and how often measurements are conducted. Furthermore, there is a dependency on the measured water matrix as well, e.g. high turbidity coastal water requires more often cleanings than open ocean water. We can only make recommendations based on our experiences with the analyzer. During our field

deployments of the analyzer (not part of this study), we ran a cleaning procedure using ethanol right before a new "calibration" of the system with CRM. As our analyzer measured around 1000 $A_T$ values per month, we carried out an ethanol flush with a subsequent calibration on monthly basis. We also experienced that the subsequent drift correction is entirely manageable up to a maximum $\Delta A_T$ of approximately 30 $\mu mol\,kg^{-1}$ (as observed during our field deployments, not part of this study). This $\Delta A_{T,max}$ can be used as a guidance for determining the cleaning frequency.

Another important outcome is that the shape of the drift differed with the amount of impurities. Below a certain purity grade (between 93.2 % and 98.1 %), the drift behavior appears to change from linear to non-linear. However, for unattended long-term installations of the CONTROS HydroFIA® TA analyzer it is highly preferable to have a linear drift. Under this condition, the correction during the post-processing of the data is easier and the necessary reference measurements can be reduced to a pre- and post-deployment measurement. Furthermore, the upper limit of the analyzer's working range will be reached faster with a non-linear increase of the $A_T$ values per measurement. Hence, there is the risk, that the measured $A_T$ values are rendered useless towards the end of a long-term deployment.

Due to the nearly similar drift behavior of Acros Organics and Alfa Aesar BCG, we also hypothesize that the observed non-linear behavior was mainly caused by the impurity with the retention time around 58 min, which is only present in BCG from Acros Organics, Carl Roth, and Alfa Aesar. Additional tests with the Carl Roth indicator supported the hypothesis (results not shown). This certain impurity might be a molecule with a higher adsorption tendency to the glass wall of the cuvette compared to other impurities. If the used indicator dye contains this impurity type, the magnitude and shape of the drift is mainly driven by the presence of this molecule than by the BCG purity itself. As a consequence, the usage of BCG indicators containing this impurity should be avoided especially during long-term deployments.

Additional to the impacts on the drift, we also experienced, that the frequency of system cleanings has to be increased when using BCG with low purity. For unattended long-term deployments, this must be taken into account.

### 3.2.2 Standard addition experiment

After this experiment was conducted, we experienced that "high-purity" ($P_{BCG} > 98\,\%$) and "low-purity" BCG ($P_{BCG} < 94\,\%$) showed different results. Hence, we decided to divide the results and discussion section of this experiment into two groups. Which type of BCG belongs to which group can be found in Table 4. We cannot say anything about the behavior of BCG with $98\,\% > P_{BCG} > 94\,\%$, because none of the tested dyes fall in this range.

The results of the standard addition experiment carried out with purified and unpurified TCI BCG ("high-purity" BCG) are shown in Fig. 5. By plotting $A_{T,measured}$ vs. $A_{T,titrated}$, purified and unpurified TCI BCG show a linear equation of $y = (0.996 \pm 0.013) \times x + (11 \pm 29)$, and $y = (0.997 \pm 0.012) \times x + (7 \pm 26)$, respectively. Both correlations satisfy the quality requirements (slope = 1, intercept = 0) within their uncertainty, and they were statistically indistinguishable. Hence, the sensitivity and linearity of these measurements are considered satisfactory. The evaluation of precision and bias, which is summarized in Table 4 revealed no significant differences between measurements. Furthermore, both biases were in full agreement with previous laboratory performance characterizations of the system (Seelmann et al. (2019): $\pm\,1.0\,\mu mol\,kg^{-1}$) and with the requirements of Dickson et al. (2007) for standard open-cell $A_T$ titrators for which an overall bias of $\pm\,2\,\mu mol\,kg^{-1}$ is required.

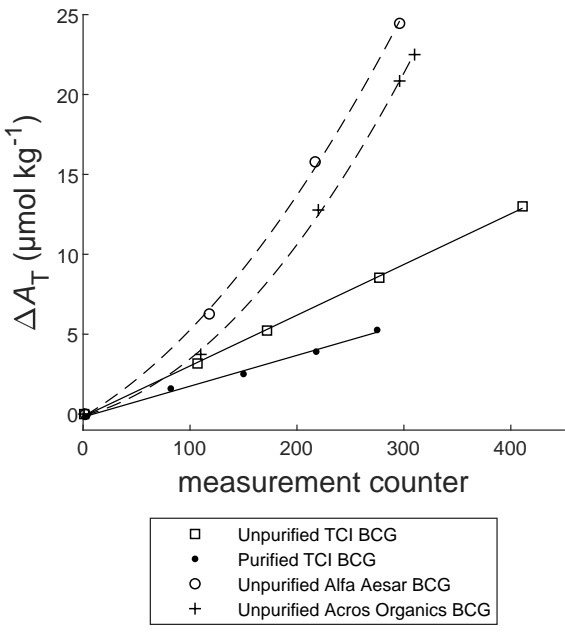

**Figure 4.** Bias ($\Delta A_{\mathrm{T}}$) between measured $A_{\mathrm{T}}$ and reference value of the CRM as a function of the measurement counter of the CONTROS HydroFIA® TA analyzer, where filled circles, open squares, crosses, and open circles represent the average of five repeated measurements made with purified BCG (TCI), unpurified TCI BCG, unpurified Acros Organics BCG, and unpurified Alfa Aesar BCG, respectively. The solid lines are the linear regressions of the associated measurement points. The dashed lines represent a non-linear regression.

However, the requirement for precision (standard open-cell titrator: $\pm\,1\,\mu\mathrm{mol}\,\mathrm{kg}^{-1}$) were not fully achieved, but both results are entirely comparable to our previous laboratory performance characteristic (Seelmann et al. (2019): $\pm\,1.5\,\mu\mathrm{mol}\,\mathrm{kg}^{-1}$). Con-
sequently, above a relative purity grade of 98 % no negative influence of indicator impurities on the measurement performance
of the analyzer could be identified.

"Low-purity" indicators behaved completely different. The results of the standard addition experiment carried out with
unpurified BCG from Acros Organics and Alfa Aesar are shown in Fig. 5 with their linear equations of $y = (1.097 \pm 0.013) \times$
$x + (-228 \pm 29)$, and $y = (1.147 \pm 0.036) \times x + (-352 \pm 82)$, respectively. Clearly, these correlations were not satisfactory,
and statistically different to the correlation of purified BCG. Hence, these "low-purity" dyes do not show the sensitivity and
linearity behavior that is required for most accurate measurements with the analyzer. Table 4 shows that measurements with
Acros Organics BCG ($P_{\mathrm{BCG}}$ = 93.2 %) still fell within acceptable ranges regarding precision and bias requirements. However,
measurements using Alfa Aesar BCG ($P_{\mathrm{BCG}}$ = 85.4 %) did not meet the quality requirements.

Summing up, we can state that the uncertainty of $A_{\mathrm{T}}$ measurements only deteriorates significantly for a BCG purity grade
below 94 %. Indicator dyes with $P_{\mathrm{BCG}} > 98$ % provide $A_{\mathrm{T}}$ measurements with a quality comparable to these measured with

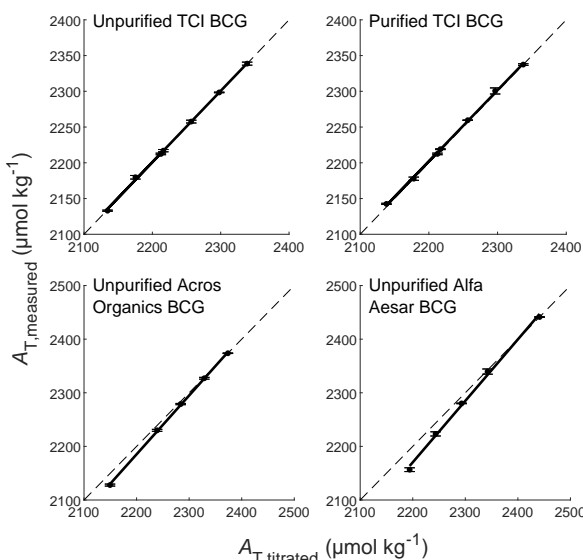

**Figure 5.** $A_{\mathrm{T,measured}}$ as a function of $A_{\mathrm{T,titrated}}$ of each titration step measured with purified and unpurified TCI BCG as well as unpurified BCG from Acros Organics and Alfa Aesar. The black filled circles represent the average of five repeated measurements for each sample with their standard deviations ($\sigma$) as errorbar. The black solid lines indicate the linear fit of the data points. The black dashed lines indicate the 1-to-1 line of these plots.

255 purified BCG. These findings partially support other studies dealing with different purified pH indicators for spectrophotometric pH (Yao et al., 2007; Liu et al., 2011; DeGrandpre et al., 2014; Lai et al., 2016) and $A_{\mathrm{T}}$ measurements (Nand and Ellwood, 2018). There, indicator purification always led to an improvement in measurement precision. Under the scope of this study, we proved that purification of BCG is not necessary to improve the quality of the $A_{\mathrm{T}}$ measurements with the CONTROS HydroFIA® TA analyzer as long as the used BCG is purer than 98 %. The reason for the deviation to other studies lies

260 in the measuring principle of the system. Before starting measurements with newly prepared solutions, it is obligatory to "calibrate" the system by measuring a CRM. However, this procedure is not a calibration in the real sense, as the method has an absolute character. During this routine, the exact volume of the analyzers internal seawater sample loop, $V_{\mathrm{SW}}$ is determined being the only unknown variable within this method. Hence, all inevitable uncertainties (including impurities of the indicator) are combined in $V_{\mathrm{SW}}$ and thereby taken into account for subsequent $A_{\mathrm{T}}$ measurements. The present results prove that this

265 procedure is able to compensate any influences of indicator impurities on the measurement quality up to an impurity level of 2 %. Consequently, the usage of "low-purity" BCG is not recommended.

**Table 4.** Precision and bias of unpurified and purified BCG

| | "High-purity" BCG: | | "Low-purity" BCG: | | Requirements | Typical |
|---|---|---|---|---|---|---|
| | unpurified TCI | purified TCI | unpurified Acros Organics | unpurified Alfa Aesar | for standard open-cell titrators[a] | performance of the analyzer[b] |
| Precision $\sigma$ ($\mu$mol kg$^{-1}$) | $\pm$ 1.6 | $\pm$ 1.5 | $\pm$ 1.4 | $\pm$ 2.7 | $\pm$ 1 | $\pm$ 1.5 |
| Bias (RMSE) ($\mu$mol kg$^{-1}$) | $\pm$ 1.2 | $\pm$ 1.1 | $\pm$ 1.7 | $\pm$ 5.3 | $\pm$ 2 | $\pm$ 1.0 |

[a]Dickson et al. (2007); [b]Seelmann et al. (2019)

## 4 Cost-benefit analysis

### 4.1 Measurements with purified vs. unpurified BCG

This study proves that an HPLC purification of BCG is entirely feasible. But is the purification of BCG worths the efforts and costs involved? To answer this question we compare the costs incurred and the benefits gained for $A_T$ measurements with the CONTROS HydroFIA® TA analyzer. Due to the relatively long HPLC run time of 60 min and a flow rate of 31.2 mL min$^{-1}$, the purification method needs about 1.5 L of ACN per run (including pre- and post-flushes). To carry out the long-term and standard addition experiment for this study (around 500 measurements), approximately 144 mg of purified BCG were needed. Hence, with a yield of around 50 mg pure BCG per purification run, a minimum of three injections was necessary. However, for long-term measurement campaigns with the analyzer, the typical volume of BCG solution is 500 mL, which is sufficient for at least 2300 measurements. This would need 700 mg of purified BCG, which corresponds to a minimum of 14 purification runs and 21 L of ACN. ACN with HPLC grade is a relatively expensive chemical, and it must be appropriately disposed. This causes additional costs. Furthermore, the whole purification process takes about a full working day per run.

Rough calculations on the actual costs per measurement with the CONTROS HydroFIA® TA analyzer revealed that indicator purification would approximately double measurement costs. The calculation for measurements with unpurified indicator are based on ready-to-use 500 mL cartridges for both chemicals (HCl and BCG) ordered from Kongsberg Maritime Contros GmbH without any preparation effort for the user.

To overcome this high increase in measurement costs, there could be the possibility to develop a flash chromatography (FC) purification method for BCG as described for mCP by Patsavas et al. (2013a) to increase the yield of purified dye per run. According to the method description in this publication (Patsavas et al., 2013a), the solvent consumption of both methods (FC and HPLC) per purification run is approximately the same. Provided the FC method would increase the yield 3.5 times as it was described for mCP, only 4 injections will be necessary to produce enough purified BCG for a long-term deployment with at least 2300 measurements. The estimated measurement costs for such a FC method would be approximately a third of these for measurements using BCG purified by HPLC. Hence, if BCG purification would be necessary, the FC method

would be the more cost-effective choice. However, it has to be taken into account that the calculations for these measurement costs (especially for the FC method) are just theoretically estimated and may differ from reality depending on availability of resources and equipment. Furthermore, a FC purification method for BCG is so far not developed and validated, which means additional costs and working time.

Finally, if we compare the purified BCG with "high-purity" BCG like from TCI, the only benefit gained from the purification is a reduced drift per $A_T$ measurement. However, as long as the drift pattern is linear, its actual slope is irrelevant as it can be easily corrected by regular reference measurements. Furthermore, there is no improvement in the measurement quality (precision and bias) as long as the impurity level is 2 % or below. Since the drift behavior cannot fully overcome, it seems not worth the effort to purify BCG for $A_T$ measurements with the CONTROS HydroFIA® TA analyzer.

The types and quantities of impurities can nevertheless have a strong influence on measurement quality in unattended long-term applications of the system as it was shown before (e.g. change of the drift behavior, non fulfillment of the quality requirements). Hence, the purity of the used BCG is not unimportant at all. To achieve the best long-term measurement experience with the analyzer it is not necessary to use purified BCG, as the purest available indicator (e.g. BCG from TCI) generate fully satisfying quality results. Users of the CONTROS HydroFIA® TA should take the consequences of indicator impurities into account when choosing their BCG supplier. From this perspective, it would be beneficial to invest into higher purity indicator avoiding the issues described above. If applicable, an HPLC analysis of the used indicator following the here described analytical method can show any types and quantities of impurities. However, if there is no HPLC available, long-term laboratory measurements as described here can help to evaluate whether the purchased indicator is suitable or not by evaluating the drift behavior. As there could be batch to batch variability in purity, the drift pattern should be also assessed for each batch of BCG provided by the same supplier.

## 4.2 BCG characterization

Most of the studies dealing with purification of indicator dyes for spectrophotometric seawater pH measurements conducted a subsequent characterization of the purified indicator (e.g. Liu et al., 2015; Patsavas et al., 2013b; Nand and Ellwood, 2018). Due to impurity impacts, coefficients and constants of purified indicators may be different to these of unpurified dyes. During our work with purified BCG, we decided to forgo of an indicator characterization. There were two reasons for this decision:

1. Li et al. (2013) investigated the impact of different BCG characteristics found in the literature on spectrophotometric $A_T$ measurements and concluded that the influences are insignificant also with regard to possible impurities. They justified this conclusion with the calibration of the system using CRM. The CONTROS HydroFIA® TA analyzer follows a similar measurement principle as the analyzer described by Li et al. (2013) and also conducted a calibration routine. Therefore, any uncertainties regarding the coefficients are taken into account for subsequent measurements.

2. The measurement quality using both purified and unpurified "high-purity" BCG were fully satisfying and met the quality requirements for $A_T$ measurements. Furthermore, both uncertainties did not significantly differ from each other.

Finally, we concluded that a characterization of purified BCG would not improve the measurement quality at all and therefore decided to not conduct it.

## 5 Conclusions

We successfully developed an HPLC purification method for BCG and subsequently tested the impact of using the purified and unpurified dye on measurements with a novel autonomous analyzer for seawater $A_T$, the CONTROS HydroFIA® TA.

Taking all the achieved results into account, we conclude that a purification of BCG is not strictly recommended to carry out high-quality measurements with the CONTROS HydroFIA® TA analyzer. But the usage of "high-purity" BCG ($P_{BCG} >$ 98 %, e.g. from TCI) is highly recommended to avoid a non-linear drift behavior and resulting loss of measurement quality as it was observed with "low-purity" BCG ($P_{BCG} < 94$ %). Users of the CONTROS HydroFIA® TA analyzer should take these recommendations into account, if they want to prepare the BCG solution on their own. A preceding HPLC analysis of the indicator dye would be the preferred approach to test the BCG purity and avoid a loss of analytical performance. BCG indicator dyes showing a relatively big peak after the BCG peak should be avoided, because their usage results in a non-linear drift pattern. It must be noted that modified HPLC methods (e.g. with a different mobile phase composition, detector or column) may result in altered peak patterns or relative BCG purities. However, not every user of the CONTROS HydroFIA® TA has the facilities for such HPLC analyses. In case of any doubts, the compatibility of the purchased BCG can be easily tested by applying the laboratory long-term measurement experiment explained in this study. Dyes resulting in a linear drift pattern can be used without any concern providing the cleaning intervals are performed regularly to limit the absolute drift to $< 30\ \mu\mathrm{mol\,kg^{-1}}$. Furthermore, the measurement quality should be monitored on regular basis, especially if the BCG solution decomposes over time. These tests should be also conducted with a new batch of BCG from the same vendor, because there could be batch-to-batch variabilities in purity.

*Author contributions.* KS: Conceptualization, Methodology, Validation, Formal analysis, Investigation, Writing - Original Draft, Visualization; MG: Methodology, Resources, Writing - Review & Editing; SA: Conceptualization, Methodology, Writing - Review & Editing; AK: Conceptualization, Resources, Writing - Review & Editing, Supervision, Funding acquisition

*Competing interests.* At the time of this study the co-author S. Aßmann was an employee of Kongsberg Maritime Contros GmbH (Kiel, Germany), which commercialized the used analyzer. However, its measurement principle follows conceptual design and dedicated studies performed during his doctorate in our working group. Apart from this there are no conflicts of interests to declare.

*Acknowledgements.* This work was supported by the European Union's Horizon 2020 Research and Innovation Programme (AtlantOS) (grant number: 633211).

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
