# Peer review of "Impact of impurities in bromocresol green indicator dye on spectrophotometric total alkalinity measurements"

_Ocean Science, 2019_

## Referee Comment (RC1) · Anonymous Referee #1 · 17 Jan 2020

Referees Comments

Firstly, the paper justifies the effort on the assumption that impurities in BCG indicator impact spectrophotometric total alkalinity measurements. It seems to be good paper. Three things were true that led to spectrophotometric pH requiring BCG purification are the impurities in dyes, the impurities cause drift in total alkalinity for the system used for the novel autonomous analyser CONTROS HydroFIAr TA and lastly no two sets of dye had the same impurities.

On a similar note, the authors claim that BCG with impurity quantities higher than 6 % provided AT values, which failed fundamental quality requirements but still conclude

that to gain optimal AT measurements, an Indicator purification is not necessarily required as long as the purchased dye has a purity level of at least 98 % and they are able to provide quality measurements to avoid identified issues. I don't see how this is true. Purification of dye is expensive but then it is not strictly recommended by the author to carryout high quality measurements. I guess with high quality measurements nothing should be compromised. I think there are benefits to this approach, but the authors need to be clearer and accurately spell out what they are as stated in abstract (line 6-7) that impurities and quality of impurities do impact the drift behaviour of the analyser. So my question is that how accurate are these total alkalinity measurements using the analyser and are they taken into account when the total alkalinity is determined.

Second, assessment: I have concerns that characterisation of the pure BCG and impure BCG would results in separate values for the extinction coefficients. I don't see any section in paper that shows the characterisation of pure BCG was conducted.

If the paper is accepted for publication, I hope the authors could make their points clear so the reader could make proper decision for their research needs.

There are typos in the manuscript which I feel needs to be restructured. Specifically Line 3 influences from impurities. I believe it should read as influences of impurities.

Line 8: Could you please specify the kind of drift. Whether there is change in total alkalinity or how the drift is caused by the impurities. Lines 40 describe to described.

Section 2.2 It is stated that all analysis were carried out in air conditioned labs. My question is based on the temperature range for the instrument and sample what was the approximate temperature conditions. As I believe that most of the indicators have a temperature range where they are most effective and work the best.

Line 106-107 was the purified BCG prepared using the sodium salt in order to make sure that samples and indicators are of similar ionic strength? Line 120 Equation 2 shows how the precision was calculated. It would be nice to show in the form of an

equation how the total alkalinity for the samples was calculated as well.

Line 144 additionally could be changed to 'in addition to'

Line 178 reset to 'resets'. What is the frequency of cleaning the analyser? Could you specify please. And are standard runs or CRM used in between runs to maintain the calibration.

Line 189 to 191. There is something wrong and it is difficult to understand. Probably reword or restructure the sentences so that it is easy to understand. I don't understand how the characters the author is mentioning in this section. Was pure BCG characterised?

Line 199 Figure 5 appears on page 12. Could it be moved closer to where it is mentioned in the text for easier referral?

Line 204 Author refers to paper by Seelmann et al., 2019 and refers to accuracies I when compared to CRM. It would be nice to at least state some values here so that it is easier for the readers to follow though.

Lines 215 reword the sentence probably.

Line 217- 229 how the total alkalinity measurement deteriorates. This section is a bit confusing as the author tries to show total alkalinity and with that talks about the precision and accuracy. Could the author be more specific? Probably with the help of equation or something how their system compares to other studies.

Line 245 what does FC refer to in this section.

Line 250 with BCG i.e. can be changed to 'using BCG'

Lines 257 delete 'be'

---

## Referee Comment (RC2) · Truls Johannessen (Referee) · 6 Feb 2020

This paper is general well written and the presentation of results are appropriate and comprehensive, and their arguments that indicator purification increase the quality of the alkalinity measurements are well justified and summarized in figure 4 and 5.

The important issue now is: To what degree can their results be justified and used by other groups using the same instrument set up as Seelmann et al did?

The cost assessment seems to give an additional dimension to the paper. Of course, we should always strive towards simplification and better cost efficiency as long as this

don't cause a compromise to the precision and accuracy of measurements required to address scientific questions pursued. The benefit here must always be balanced by the general costs of the fieldwork campaign and costs related to manpower in use. These costs often greatly exceed the costs in performing the analytical work and then it will be better to make sure that measurements are performed the best way there is.

There is a clear recommendation and the end of the paper claiming the following lines 260 and 268:

To achieve the best long-term measurement experience with the analyzer it is not necessary to use purified BCG, as the purest available indicator (e.g. BCG from TCI) generate fully satisfying quality results. Users of the CONTROS HydroFIA TA should take the consequences of indicator impurities into account when choosing their BCG supplier. From this perspective, it would be beneficial to invest into higher purity indicator avoiding the issues described above. If applicable, an HPLC analysis of the used indicator following the here described analytical method can show any types and quantities of impurities. However, if there is no HPLC available, long-term laboratory measurements as described here can help to evaluate whether the purchased indicator is suitable or not by evaluating the drift behaviour.

This paper is a valuable contribution to the scientific community dealing with delicate measurements, in this case of the carbon system variable alkalinity. It stimulated discussions related to the use of different dye(s) and their purity.

This is convincingly stated in lines 255-257:

Finally, if we compare the purified BCG with "high-purity" BCG like from TCI, the only benefit gained from the purification is a reduced drift per AT measurement. There is no improvement in the measurement quality (precision and accuracy) as long as the impurity level is 2 % or below.

My conclusion is that this paper can be published with minor revision (typos).

[Figure]

---

## Author Comment (AC1) · 6 Mar 2020

**Reply to the comments of Anonymous Referee 1 (received on 17 Jan 2020)**

First, we thank the Anonymous Referee for the evaluation of our manuscript and constructive suggestions and comments. We answer each comment point-by-point in the following text. All manuscript changes based on these comments can be found in the supplements with marked differences to the previous version. Furthermore, all lines named in the following answers refer to the supplemented changed manuscript.

**Comment #1:** Firstly, the paper justifies the effort on the assumption that impurities in

[Figure]

BCG indicator impact spectrophotometric total alkalinity measurements. It seems to be good paper. Three things were true that led to spectrophotometric pH requiring BCG purification are the impurities in dyes, the impurities cause drift in total alkalinity for the system used for the novel autonomous analyser CONTROS HydroFIAr TA and lastly no two sets of dye had the same impurities. On a similar note, the authors claim that BCG with impurity quantities higher than 6% provided AT values, which failed fundamental quality requirements but still conclude that to gain optimal AT measurements, an Indicator purification is not necessarily required as long as the purchased dye has a purity level of at least 98 % and they are able to provide quality measurements to avoid identified issues. I don't see how this is true. Purification of dye is expensive but then it is not strictly recommended by the author to carryout high quality measurements. I guess with high quality measurements nothing should be compromised.

**Answer #1:** You are totally right by saying "with high quality measurements nothing should be compromised". But we also think that there must be a reasonable balance between the effort to achieve high-quality measurements and their costs. Most users of spectrophotometric $A_T$ analyzers (no matter if they work at research institutes, industry, or somewhere else) are dependent on a certain budget for their measurements. Additionally, they are maybe not able to perform indicator purifications because they do not have the necessary facilities at hand. Consequently, they have to use unpurified BCG as long as there is no commercial provider for purified BCG. Our results show that the quality $A_T$ measurements based on unpurified BCG is fully comparable to those made with purified BCG as long as the impurity quantity does not exceed a certain level (see Table 4). We also show that there is at least one provider of BCG who can routinely fulfill this requirement (there could be others, but during our study we were not able to figure them out). These are the reasons why we do not strictly recommend a purification of BCG to gain high-quality measurements. At the same time, however, we want the reader to be aware of the fact that the impurity level of their purchased BCG is not unimportant at all. We changed the abstract and parts of the manuscript in order to better address this (see supplement).

**Comment #2:** I think there are benefits to this approach, but the authors need to be clearer and accurately spell out what they are as stated in abstract (line 6- 7) that impurities and quality of impurities do impact the drift behaviour of the analyser.

**Answer #2:** Yes, there are benefits to this approach which are stated in lines 298 - 299 of the supplemented manuscript. But in comparison to "high-purity" BCG purchased from TCI, which can be used without any extra work, the gained benefits are not significant enough to justify the need and corresponding effort of indicator purification (it does not measurably improve the precision and accuracy of the measurements). In our opinion, there is no reason why a smaller linear drift is principally better than a somewhat larger **linear** drift as both of them are easily correctable by regular reference measurements. The only thing to avoided is a non-linear drift behavior which we found for the less pure commercial BCG products, hence no need to inform the scientific community about this issue. We changed the cost-benefit analysis (lines 299 - 300) to better address this statement. Of course, in comparison to "low-purity" BCG, purification is beneficial. But we think that the readers of this manuscript would prefer to buy a slightly more expensive unpurified BCG than have extra work with the purification (providing they have the facilities to purify BCG). In addition, we added a hint that the drift pattern should be assessed for each batch of BCG from the same supplier. This procedure ensures that batch to batch variability in purity is monitored (see lines 311 - 313).

**Comment #3:** So my question is that how accurate are these total alkalinity measurements using the analyser and are they taken into account when the total alkalinity is determined.

**Answer #3:** Our previous work with this analyzer (Seelmann et al., 2019) revealed a relative analyzer uncertainty of 0.08 % under laboratory conditions. This was determined by the same standard addition experiment as described in this work (see Sect. 2.2.3). And yes, the uncertainty of the $A_\mathrm{T}$ measurements was taken into account (manuscript was changed to address this, see lines 195 - 196).

**Comment #4:** Second, assessment: I have concerns that characterisation of the pure BCG and impure BCG would results in separate values for the extinction coefficients. I don't see any section in paper that shows the characterisation of pure BCG was conducted.

**Answer #4:** A full characterization of the purified BCG was beyond the scope of the study and therefore, not conducted. Of course, it is beyond a doubt that purified BCG might have different extinction coefficients than impure BCG (similar to mCP). In our study, all $A_\mathrm{T}$ values were calculated using the coefficients reported by Breland and Byrne (1993). However, Li et al. (2013) investigated the influence of different BCG constants and coefficients on the measured AT value and concluded that they are insignificant (also with regard to indicator impurities). The reason for that is the calibration of the system with CRM as it was also done during our measurements (the measurement principle of the CONTROS HydroFIA$^®$ TA follows a similar procedure like it is described by Li et al. (2013)). Therefore, any uncertainties regarding the coefficients are taken into account for subsequent measurements. And the measurement quality both with purified and unpurified "high-purity" BCG is entirely sufficient for a spectrophotometric analyzer also without a characterization. However, by using "low-purity" BCG, this calibration seems to reach its limits as the measurement uncertainty impairs. That is why we added the statement "the usage of "low-purity" BCG is not recommended" (see line 265). Otherwise a characterization of "low-purity" BCG instead of purified BCG would be necessary. And this procedure seems to be senseless, because there is no need of using "low-purity" BCG as "high-purity" BCG is available. Finally, we concluded that a characterization was not necessary. We included the section "BCG characterization" in the "Cost-benefit analysis" section (line 314 - 327) to justify our decision.

**Comment #5:** If the paper is accepted for publication, I hope the authors could make their points clear so the reader could make proper decision for their research needs.

**Answer #5:** We hope that we could clarified all unclear points named by Referee with
our changes.

**Comment #6:** There are typos in the manuscript which I feel needs to be restructured. Specifically Line 3 influences from impurities. I believe it should read as influences of impurities.

**Answer #6:** Changed (see line 3)

**Comment #7:** Line 8: Could you please specify the kind of drift. Whether there is change in total alkalinity or how the drift is caused by the impurities.

**Answer #7:** Added this information (see lines 10 – 12 and 184 - 187)

**Comment #8:** Lines 40 describe to described

**Answer #8:** Changed (see line 45)

**Comment #9:** Section 2.2 It is stated that all analysis were carried out in air conditioned labs. My question is based on the temperature range for the instrument and sample what was the approximate temperature conditions. As I believe that most of the indicators have a temperature range where they are most effective and work the best.

**Answer #9:** The CONTROS HydroFIA$^®$ TA has an internal sample temperature control where the measured sample is constantly temperature controlled to $25.0 \pm 0.1$ °C which is the ideal temperature for BCG (the used characteristics by Breland and Byrne (1993) were carried out at 25°C). The temperature control is realized by a Peltier element and temperature measurement directly behind the cuvette. Therefore, the temperature of the seawater sample is independent of the room temperature. This information has now been added to the manuscript (see line 104).

**Comment #10:** Line 106-107 was the purified BCG prepared using the sodium salt in order to make sure that samples and indicators are of similar ionic strength?

**Answer #10:** Yes, we prepared the purified BCG solution in deionized water and added

a certain amount of sodium hydroxide solution to transfer the BCG into the sodium salt (in solution). Information to this were added in the manuscript (lines 123 - 129).

**Comment #11:** Line 120 Equation 2 shows how the precision was calculated. It would be nice to show in the form of an equation how the total alkalinity for the samples was calculated as well.

**Answer #11:** Equation 2 (now Equ. 4) does not show how the precision is calculated. It shows how the accuracy is calculated. See lines 142 – 145 ("The root mean square error (RMSE) ... gave us information about the measuring accuracy. It was calculated by..."). The precision was calculated by the standard deviation. However, we decided against an equation for the standard deviation as this is a common statistical tool. General calculation equations for $A_T$ were added to the manuscript (see lines 105 - 114).

**Comment #12:** Line 144 additionally could be changed to 'in addition to'

**Answer #12:** Changed (see lines 166)

**Comment #13:** Line 178 reset to 'resets'.

**Answer #13:** Changed (see lines 203)

**Comment #14:** What is the frequency of cleaning the analyser? Could you specify please. And are standard runs or CRM used in between runs to maintain the calibration.

**Answer #14:** The frequency of cleanings depends on the application of the analyzer and cannot be stated in detail. For example: the more measurements are carried out with the analyzer the higher is the frequency of cleanings. That of course differs from user to user. Furthermore, there is a dependency on the measured water matrix as well, e.g. high turbidity coastal water requires more often cleanings. We made a recommendation based on our experiences (see lines 204 - 210)

[Figure]

**Comment #15:** Line 189 to 191. There is something wrong and it is difficult to understand. Probably reword or restructure the sentences so that it is easy to understand. I don't understand how the characters the author is mentioning in this section.

**Answer #15:** Reworded the sentences for a hopefully better understanding (see lines 223 - 226)

**Comment #16:** Was pure BCG characterised?

**Answer #16:** No (please see Answer 4)

**Comment #17:** Line 199 Figure 5 appears on page 12. Could it be moved closer to where it is mentioned in the text for easier referral?

**Answer #17:** Unfortunately, the placement of the figures is automatically performed by LaTeX, which was used for preparing the manuscript (using the Copernicus LaTeX template). We tried to move it, but it was not possible. Unfortunately, the movement of the figure is shown as a change in the manuscript by using the tool "latexdiff" (see caption of Fig. 5 and lines 266 – 269). Please ignore these changes. However, we think that the placement of the figures will be improved during the typesetting progress.

**Comment #18:** Line 204 Author refers to paper by Seelmann et al., 2019 and refers to accuracies I when compared to CRM. It would be nice to at least state some values here so that it is easier for the readers to follow though.

**Answer #18:** Added (see lines 239 – 240 and 243). We also added the requirements and the typical performance of the analyzer in Table 4 for a better comparison.

**Comment #19:** Lines 215 reword the sentence probably.

**Answer #19:** Reworded the sentence (see lines 249 - 252)

**Comment #20:** Line 217- 229 how the total alkalinity measurement deteriorates. This section is a bit confusing as the author tries to show total alkalinity and with that talks about the precision and accuracy. Could the author be more specific? Probably with

the help of equation or something how their system compares to other studies.

**Answer #20:** Please see Table 4 (page 13) in the changed manuscript (changed because of a previous comment). There is a comparison of the measurement uncertainties for purified, "high-purity" and "low-purity" BCG. Additionally, the fundamental $A_\mathrm{T}$ quality requirements reported by Dickson et al. (2007) can be found there in addition to the typical performance of the analyzer found during our previous study (Seelmann et al., 2019). It is obvious that the measurement quality (especially accuracy) deteriorates with increasing impurity level.

**Comment #21:** Line 245 what does FC refer to in this section.

**Answer #21:** FC means flash chromatography. This abbreviation was firstly explained in the Introduction section. But we added an additional explanation there (see line 287)

**Comment #22:** Line 250 with BCG i.e. can be changed to 'using BCG'

**Answer #22:** Changed (see line 293)

**Comment #23:** Lines 257 delete 'be'

**Answer #23:** Deleted (see line 301)

Please also note the supplement to this comment:
https://www.ocean-sci-discuss.net/os-2019-126/os-2019-126-AC1-supplement.pdf

**Supplement:**

[revised manuscript text omitted]

---

## Author Comment (AC2) · 6 Mar 2020

**Reply to the comments of Truls Johannessen (Referee #2) (received on 06 Feb 2020)**

First, we thank Truls Johannessen for the evaluation of our manuscript and his highly supportive comments. We answer each comment point-by-point in the following text.

**Comment #1:** This paper is general well written and the presentation of results are appropriate and comprehensive, and their arguments that indicator purification increase the quality of the alkalinity measurements are well justified and summarized in figure 4

and 5.

**Answer #1:** Thank you for this supportive evaluation of our manuscript. We appreciate it.

**Comment #2:** The important issue now is: To what degree can their results be justified and used by other groups using the same instrument set up as Seelmann et al did? The cost assessment seems to give an additional dimension to the paper. Of course, we should always strive towards simplification and better cost efficiency as long as this don't cause a compromise to the precision and accuracy of measurements required to address scientific questions pursued. The benefit here must always be balanced by the general costs of the fieldwork campaign and costs related to manpower in use. These costs often greatly exceed the costs in performing the analytical work and then it will be better to make sure that measurements are performed the best way there is. There is a clear recommendation and the end of the paper claiming the following lines 260 and 268: "To achieve the best long-term measurement experience with the analyzer it is not necessary to use purified BCG, as the purest available indicator (e.g. BCG from TCI) generate fully satisfying quality results. Users of the CONTROS HydroFIA TA should take the consequences of indicator impurities into account when choosing their BCG supplier. From this perspective, it would be beneficial to invest into higher purity indicator avoiding the issues described above. If applicable, an HPLC analysis of the used indicator following the here described analytical method can show any types and quantities of impurities. However, if there is no HPLC available, long-term laboratory measurements as described here can help to evaluate whether the purchased indicator is suitable or not by evaluating the drift behaviour." This paper is a valuable contribution to the scientific community dealing with delicate measurements, in this case of the carbon system variable alkalinity. It stimulated discussions related to the use of different dye(s) and their purity. This is convincingly stated in lines 255-257: "Finally, if we compare the purified BCG with "high-purity" BCG like from TCI, the only benefit gained from the purification is a reduced drift per AT measurement. There is

**OSD**

no improvement in the measurement quality (precision and accuracy) as long as the impurity level is 2 % or below."

**Answer #2:** Thank you for this supportive conclusion of our paper. It shows us that we will contribute beneficial information to the scientific community by publishing our results.

**Comment #3:** My conclusion is that this paper can be published with minor revision (typos).

**Answer #3:** We highly appreciate this evaluation of our manuscript. Thank you! We will do our best to correct all typos before publication.

---

## Author Response (AR2)

**Reply to comments of Mario Hoppema (Topic Editor), received on 12 Mar 2020**

First, we thank Mario Hoppema for serving as Topic Editor and for the thorough evaluation of our manuscript. Thank you for all these valuable comments that highly improve our manuscript.
We answer each comment point-by-point in the following text. A revised manuscript with marked changes based on these comments can be found later in this document. Furthermore, all lines named in this answer refer to this changed and marked manuscript.

**Comment #1:** P1, L3 „The work described here focuses on impacts of . . . " Work cannot focus on something, but you yourself can. Please correct the sentence.
**Answer # 1:** Modified the sentence (see line 3)

**Comment #2:** P1, L6 did (instead of does). Appears to fit better in such a sentence. If you need to use "does", the sentence ending must be changed.
**Answer #2:** Changed into "did" (see line 6)

**Comment #3:** P1, L6 I do not see the use of the word "subsequent" here.
**Answer #3:** The word "subsequent" refers to the development of the method, which had to be done before the analysis. But for the whole understanding of the abstract, it is not necessary. Consequently, we leave it out (see line 6).

**Comment #4:** P1, L11 "However, it could not be fully overcome." What could not be overcome? Please clarify in the text.
**Answer #4:** The "it" means the drift behavior of the analyzer. We clarified this in the text (see line 11).

**Comment #5:** P1, L19-20 the purest BCG that is purchasable.
**Answer #5:** Changed from "which" to "that" (see line 20).

**Comment #6:** P1, L22-23 "not only the biogeochemical processes but also the uptake, transport and accumulation of anthropogenic CO2 in the ocean" I think these three processes could all be taken as biogeochemical processes. Please modify text.
**Answer #6:** Changed the text (see lines 23 - 24).

**Comment #7:** P1, L23 parameters or variables?
**Answer #7:** They are variables. We changed the text with refer to this (see line 24).

**Comment #8:** L25 I think "corresponding" is not appropriate here.
**Answer #8:** As "corresponding" is not necessary for the understanding of the sentence, we leave it out (see line 25).

**Comment #9:** L27 pH measurements have been common all the time. I think "prominent" fits better here.
**Answer #9:** Modified the sentence by using "prominent" (see line 28).

**Comment #10:** L28 I think you mean "decades" instead of "centuries". Ocean carbon observations are not that old.
**Comment #10:** Yes, we meant "decades". Changed the word (see line 29).

**Comment #11:** L38 "have revealed", not reveals
**Answer #11:** Changed into "have revealed" (see line 39).

**Comment #12:** L64 please define LC here
**Answer #12:** A definition for "LC" has been included (see line 64).

**Comment #13:** L119 Would it be possible to use a different word than "calibrated", since you do not mean calibrated here, do you? Or possibly rewrite the sentence completely.
**Answer #13:** You are right, it is not a real calibration. It is a sample volume determination for the inner tubing of the analyzer, as this is the only unknown variable within the whole $A_T$ determination with the HydroFIA TA. We added a proper explanation for "calibrated" (see lines 121 - 123).

**Comment #14:** L128 accounted for instead of considered?
**Answer #14:** Changed "considered" to "accounted for" (see line 131).

**Comment #15:** L159 delete: There,
**Answer #15:** Deleted (see line 160).

**Comment #16:** L160 delete: However, (there is no contradiction here)
**Answer #16:** Deleted (see line 164).

**Comment #17:** L173 delete: respectively
**Answer #17:** Deleted (see line 177).

**Comment #18:** L194-195 This sentence is hard to read. I suggest to change it to: All AT measurements took into account the relative uncertainty of the analyzer, determined as 0.08 % (Seelmann et al., 2019).
**Answer #18:** Changed the sentence following the suggestion (see line 198 - 199).

**Comment #19:** L201 Many sentences here begin with "However". Maybe change this one?
**Answer #19:** Rearranged the sentence without "However" (see line 206).

**Comment #20:** L203-204 Maybe the authors can indicate what they consider to be

acceptable for a drifting and changing value of AT. This would give some guidance to readers/users of the system.

**Answer #20:** We added information to this (see lines 214 - 217).

**Comment #21:** L209 change to: on a monthly basis
**Answer #21:** Changed (see line 214).

**Comment #22:** L227 The phrase "For better presentation" reads odd. It is not just a matter of presentation that you are performing this. Please rephrase.
**Answer #22:** The sentence was rephrased (see line 235 - 238).

**Comment #23:** L228 delete: At the time of this study, (this is superfluous)
**Answer #23:** Deleted (see line 237).

**Comment #24:** L242 "In contrast, "low-purity" indicators behaved totally different." "in contrast" and "totally different" say exactly the same. Please rephrase.
**Answer #24:** Deleted "In contrast" as it is not necessary (see line 252).

**Comment #25:** L257 Could the authors come up with a different word than "calibration"? I think the reader might get confused by bringing it this way. If it is a real calibration, then the use of quotation marks is not necessary.
**Answer #25:** It is not a real calibration. It is an internal seawater volume determination, which is also explained before (Method section) and directly afterwards in the manuscript. The operation software of the HydroFIA calls the procedure "calibration" and we adapted this term. However, we added a statement that this is not a real calibration in order to inform the reader about this (see lines 121 - 123 and 267 - 269). We hope this is sufficient. Besides, there is exactly the same situation for the SOMMA system for DIC measurements, where the CRM measurements are essentially performed to "calibrate" the to-deliver-volume of the pipette (plus any additional smaller uncertainties such as the performance to "electron counting" in the coulometer). But this is just for your information.

**Comment #26:** Table 4 Both the precision and accuracy are given. In some cases the accuracy is better than the precision. I think this is not according to statistical principles. In the requirements for open-cell titrators according to Dickson et al 2007, one can see how it should work: the accuracy has a clearly higher absolute value (often double as large) than the precision. The reason is that for determining the accuracy, one needs to take into account the precision of the measurement of the reference material (or standard) used to determine the accuracy, and of the measurement itself. Please explain and correct your results.
**Answer #26:** We equate the bias with the accuracy (requirement of $\pm$ 2 µmol kg$^{-1}$ by Dickson et al 2007 = "overall bias" (see page 11)). The combination of accuracy and precision is then called uncertainty. But you are right, sometimes uncertainty and accuracy are the same, and a combination out of bias and precision. This depends on

the definitions. However, our "accuracy" values are the bias between the mean of the measured values and the true value (in our case the RMSE of the linear regression), as given in the Guide of Dickson et al 2007 on page 11 for standard open cell titrators: "...overall bias of less than 2 µmol kg$^{-1}$...". However, we replaced the word "accuracy" with "bias" in order to avoid confusion about the used terms (see lines 15, 147, 244, 246, 248, 258, 303 and table 4). We hope that is sufficient.

**Comment #27:** L272 appropriately, not probably
**Answer #27:** Changed (see line 283).

**Comment #28:** L294 delete However, (not necessary here, and not at the start of a paragraph)
**Answer #28:** Deleted "However" and rearranged the sentence (see line 305).

**Comment #29:** L320 delete: In this study, (superfluous)
**Answer #29:** Deleted (see line 313).

**Comment #30:** As to the Conclusions section, this is much like a summary. Although this cannot be avoided to some extent, the entire structure of this section in the present manuscript is built around this summary. See for example the last sentence of the section in L341 "Furthermore, no characterization of the purified BCG was carried out." This may well be deleted. Please restructure the Conclusions section and incorporate more real conclusions, which the manuscript definitely has to offer.
**Answer #30:** We modified the Conclusion Section to a more concluding part with recommendations for the reader/user instead of a summarizing part (see Conclusion section).

**Comment #31:** L367 CO2 with subscript
**Answer #31:** Changed (see line 389)

[revised manuscript text omitted]

$$P_{BCG} = \frac{A_{BCG}}{\sum_{i=1}^{n} A_i} \times 100\% \tag{1}$$

where $A_{BCG}$ is the area of the BCG peak, $n$ is the number of peaks, and $A_i$ is the area of the i[th] peak.

**2.1.4 Purification of BCG**

The purification was performed by the LC system in preparative mode. A 7.5 mg mL$^{-1}$ BCG solution was prepared in the mobile phase and 10 mL were injected onto the preparative column. Impurities were separated by isocratic flow (flow rate 31.2 mL min$^{-1}$) with 75:25:0.1 ACN:H$_2$O:TFA as mobile phase. The pure BCG was collected manually in a round bottom flask at its retention time of about 52 min. Approximately 90 % of the mobile phase was removed from the BCG eluate using a rotary evaporator, with the final 10 % left to evaporate in a dark open box at room temperature. The pure crystalline dye was transferred to a brown flask for further experiments.

In order to verify the success of the purification, the purified BCG was analyzed by the analytical HPLC procedure as described in Sect. 2.1.2.

**2.2 Total alkalinity measurements**

**2.2.1 Reagents and instrumentation**

Total alkalinity measurements were performed using the novel autonomous analyzer CONTROS HydroFIA$^{\circledR}$ TA (Kongsberg Maritime Contros GmbH, Kiel, Germany). Its measurement principle is based on a single-point open-cell titration of the seawater sample with subsequent spectrophotometric pH detection using BCG as indicator (Breland and Byrne, 1993; Yao and Byrne, 1998; Li et al., 2013; Seelmann et al., 2019). The seawater sample was titrated with 0.1 mol kg$^{-1}$ hydrochloric acid (HCl) obtained from Carl Roth and constantly temperature controlled to $25.0 \pm 0.1$ °C by the systems internal heat exchanger.

The $A_T$ value of the sample was calculated by the following general equation:

$$\frac{-V_{sw} \times \rho_{sw} \times A_T + V_t \times \rho_t \times C_t}{V_{sw} \times \rho_{sw} + V_t \times \rho_t} = [H^+]_F + [HF] + [HSO_4^-] + [
[revised manuscript text omitted]